# Multi-Anticancer Activities of Phytoestrogens in Human Osteosarcoma

**DOI:** 10.3390/ijms241713344

**Published:** 2023-08-28

**Authors:** Alessio Cimmino, Giovanni Francesco Fasciglione, Magda Gioia, Stefano Marini, Chiara Ciaccio

**Affiliations:** Department of Clinical Sciences and Translational Medicine, University of Rome ‘Tor Vergata’, Via Montpellier 1, I-00133 Rome, Italy; alessiocimmino7@gmail.com (A.C.); fascigli@uniroma2.it (G.F.F.); magda.gioia@uniroma2.it (M.G.); stefano.marini@uniroma2.it (S.M.)

**Keywords:** osteosarcoma, phytoestrogens, anticancer effects, estrogen receptors

## Abstract

Phytoestrogens are plant-derived bioactive compounds with estrogen-like properties. Their potential health benefits, especially in cancer prevention and treatment, have been a subject of considerable research in the past decade. Phytoestrogens exert their effects, at least in part, through interactions with estrogen receptors (ERs), mimicking or inhibiting the actions of natural estrogens. Recently, there has been growing interest in exploring the impact of phytoestrogens on osteosarcoma (OS), a type of bone malignancy that primarily affects children and young adults and is currently presenting limited treatment options. Considering the critical role of the estrogen/ERs axis in bone development and growth, the modulation of ERs has emerged as a highly promising approach in the treatment of OS. This review provides an extensive overview of current literature on the effects of phytoestrogens on human OS models. It delves into the multiple mechanisms through which these molecules regulate the cell cycle, apoptosis, and key pathways implicated in the growth and progression of OS, including ER signaling. Moreover, potential interactions between phytoestrogens and conventional chemotherapy agents commonly used in OS treatment will be examined. Understanding the impact of these compounds in OS holds great promise for developing novel therapeutic approaches that can augment current OS treatment modalities.

## 1. Introduction

Osteosarcoma (OS) is the most primary bone tumor and a major cause of cancer death during the second and third decades of life with a worldwide incidence of 3.4 cases per million people per year, for all races and both genders [1,2]. Current disease management strategies include the surgical resection of all clinically visible tumors and systemic front-line chemotherapy which uses high doses of methotrexate (MTX), cisplatin (CDDP), and doxorubicin (DOX). This determines an overall survival level of 65–70% at 5 years [3,4]. However, today, several patients continue to develop metastases with an elective site in the lung, which causes a high mortality rate. This means that 20–30% of patients are refractory to these conventional treatments [5]. This discomforting scenario is frequently attributed to the ineffectiveness of chemotherapy, which can be influenced by chemo-resistance phenomena [6]. In addition, chemotherapeutic agents often produce various side effects, including cardiotoxicity, hepatotoxicity, and renal toxicity, which contribute to the increased likelihood of OS recurrence and progression [7].

While there are an increasing number of targeted therapies being developed and an improvement in the survival rate of other cancers, OS still stands where it was decades ago. These unsatisfactory results mean that complementary and alternative treatment options merit more attention. In this regard, dietary supplements and phytotherapeutic agents, with high anticancer efficacy and nominal toxicity to normal tissues, have emerged as a promising avenue that is worth investigating [8].

Over the past decade, significant attention has been devoted to studying phytoestrogens through mechanistic in vitro research which, together with epidemiological observations, has provided evidence that supports their chemopreventive and chemotherapeutic effects in different types of malignancies, including breast, prostate, and colon cancer [8]. Phytoestrogens are found in various plants, especially soy and soy-based foods and herbal medicines [9]. They possess a structural similarity to the estrogen hormone, which enables them to interact with ERs in the body, exerting both estrogenic and anti-estrogenic effects [10,11,12]. This characteristic has sparked interest in gaining a better understanding of the potential influence of this class of plant substances on OS, considering that estrogen signaling can be implicated in the growth and progression of this malignancy [12,13,14]. In recent years, studies have shed light on the multifaceted effects of phytoestrogens on human OS cells. These compounds have been reported to exhibit both pro-apoptotic and anti-proliferative properties, suggesting their potential as therapeutic agents for OS [8,15]. Several underlying mechanisms have been proposed, including modulation of Ers, inhibition of angiogenesis, regulation of apoptosis-related proteins, and interference with cell signaling pathways involved in OS development, including the phosphatidylinositol 3-kinase (PI3K)/Akt pathway and the mitogen-activated protein kinase (MAPK) signaling and Wnt/β-catenin pathway [15]. Phytoestrogens also possess antioxidant and anti-inflammatory properties, which contribute to their anticancer effects by attenuating oxidative stress and inflammation-mediated processes [8,15]. 

In this work, we aim to provide a comprehensive synthesis of the existing body of literature surrounding the impact of phytoestrogens on human OS models, with particular attention being paid to the potential of these molecules as ER modulators. By examining preclinical studies and molecular investigations, we seek to gain insights into the mechanisms through which phytoestrogens exert their effects on OS cells and provide support for the future development of phytoestrogens as effective and safe agents for the therapy of OS. 

## 2. Phytoestrogens: Chemical Classification and General Aspects

Phytoestrogens are produced by plants (more than 300 various species) as secondary metabolites which play crucial roles in various plant functions, such as defense against pathogens, pigmentation and protection from UV radiation, photosynthetic stress, and reactive oxygen species [16]. The quantity of phytoestrogens produced by a plant increases significantly during extreme growing conditions [17]. The human diet is rich in plant-containing phytoestrogens (i.e., vegetables, legumes, cereals, fruits, nuts, etc.), and beverages, such as wine, cider, beer, tea, and many more. Many edible plants contain multiple classes of phytoestrogens, adding to their diversity and potential health benefits [9]. Regarding their structural features, phytoestrogens represent a large and heterogenic class of non-steroidal substances characterized by a close structural similarity to the principal mammalian estrogen 17β-estradiol (E2) [11,12].

Shared structures include a phenolic ring and a pair of hydroxyl groups in opposite positions on the molecule (as in the case of E2 molecule), which are responsible for the interaction of phytoestrogens with the ligand-binding domain of ERα and ERβ.

The exact position and number of these hydroxyl substituents is crucial in determining the binding affinity for the ERs and the activation of hormonal signaling [18,19].

The estrogenic or antiestrogenic properties of phytoestrogens in the target cells depend on their phenolic ring [20].

The phytoestrogens have been categorized into two main groups: flavonoids and non-flavonoids based on their chemical structure and properties. The classification and the basic structures of the most representative dietary phytoestrogens are illustrated in Figure 1. For further details, see Section 6 and Section 7. Isoflavones, flavones, flavonols, flavanones, catechins, and coumestans belong to flavonoids. Non-flavonoids include stilbenes and lignans. The most commonly occurring phytoestrogens are the flavonoids (of which the coumestans and isoflavones have the greatest estrogenic effects) and the lignans [8,21].

### 2.1. Flavonoids

Flavonoids are a large group of substituted phenolic compounds [22]. They share a common structure consisting of a fifteen-carbon skeleton composed of two benzene rings (A and B) connected by a heterocyclic pyran structure (C) in a C6–C3–C6 arrangement, as illustrated in Figure 1 [23]. The basic flavonoid skeleton can have numerous substituents, including hydroxyl groups typically found at positions 4′, 5′, and 7. Flavonoids can be further classified into different subclasses. Specifically, isoflavones are flavonoids, where the B ring is connected to the heterocyclic ring at the C3 position. 

On the other hand, flavonoids in which the B ring is linked at position 2 are divided into several subgroups, namely flavones, flavonols, flavanones, and catechins, depending on the degree of saturation and oxidation of the C ring (Figure 1). Coumestans are a distinct flavonoid which are characterized by a 1-benzoxolo (3,2-c)chromen-6-one structure formed by a benzoxole fused with a chromen-2-one [8]. The presence and position of hydroxyl groups and/or additional substituents contribute to the diversity of flavonoids and their biological activities [23,24,25]. 

Moreover, the addition of lipophilic prenyl side-chains can occur at different positions of the flavonoid skeleton, resulting in various prenylated derivatives with improved bioactivities and higher affinity to biological membranes. Prenylated flavonoids are much less common than flavonoids in nature [26].

Flavonoids are some of the most prevalent compounds found in fruits, vegetables, legumes, and tea and are generally concentrated in the fruit skin, bark, and flowers of plants [21]. Certain flavonoids, such as the flavonol quercetin, are found in all plant products (i.e., fruit, vegetables, cereals, leguminous plants, tea, and wine), but others are specific components of particular foods (i.e., flavanones in citrus fruit and isoflavones in soya). In most cases, food contains complex mixtures of flavonoids; for many food products, the composition is less known (for review, see [27]). In plants, most flavonoids are conjugated with one or more sugar residues linked to hydroxyl groups or aromatic carbons, so they mainly exist as glycosides [28]. 

### 2.2. Non-Flavonoids

Non-flavonoids encompass a broad range of plant compounds that do not possess the characteristic flavonoid structure. Their structure consists of phenolic acids in either C6–C1 (benzoic acid) or C6–C3 (cinnamic acid) conformations and are mainly represented by lignans and stilbenes (Figure 1). Non-flavonoids may occur in the form of aglycones and glycosides [28].

Lignans are dimers of phenylpropanoid units connected via two specific carbons (C-2–C-2′) and are typically found in plant cell walls [29]. They are widespread and their content is high in common foods, including grains, nuts, seeds, vegetables, and drinks such as tea, coffee, or wine. Plant lignans are the principal source of dietary phytoestrogens in the Western diet [30]. Compounds, such as pinoresinol, lariciresinol, sesamin, enterolactone, and enterodiol can be found in this group [29].

Stilbenes are among the most relevant non-flavonoid phytoestrogens which consist of a 1,2-diphenylethylene nucleus that generates two isomers (cis and trans), with the trans-isomer being the most stable and biologically active [31,32]. More than 400 stilbene compounds have been identified in plants, with various structures ranging from monomers to octamers with different substituents, such as glycosyl, hydroxyl, methyl, or isopropyl radicals [31]. Monomeric stilbenes have been studied the most. These include resveratrol and polydatin (Section 7). They are naturally occurring in fruits, mostly in grapes, berries, and peanuts [33]. In general, the occurrence of stilbenes in the human diet is limited but represents an important part of phytoestrogen intake by people who follow a Mediterranean diet or who regularly drink wines.

### 2.3. Metabolism of Dietary Phytoestrogens

Each class of dietary phytoestrogens has its own structural particularities, and studies regarding their bioavailability and metabolism are still far from being completed. There is no relation between the quantity of phytoestrogens in food and their bioavailability in the human body. Indeed, the rate and extent of absorption of dietary phytoestrogens in the intestine is determined primarily by their chemical structure and by factors such as molecular size and solubility, extent of glycosylation, hydroxylation, acylation, and degree of polymerization [12,34]. Most ingested phytoestrogens (e.g., isoflavones, lignans, and stilbenes), are predominantly present as estrogenically inactive glycosides in plant material [35]. After ingestion, phytoestrogens undergo extensive metabolization mediated both by tissue enzymes and gut microbiota, either prior to absorption or during enterohepatic circulation. The intestinal flora is capable of transforming aglycones into bioactive metabolites that are more similar to estrogens, being able to interfere with the endogenous estrogen signaling and associated cellular processes. In some cases, these metabolites have greater biological activities and different impacts on targeted tissues than their parent precursors [36,37]. For a detailed background on the absorption and metabolism of different phytoestrogens, see ref. [38]. Thus, individual variability in gut microbiota can influence the metabolism of these estrogenic molecules, contributing to their intake and beneficial effects [39]. Consequently, the identification, quantification, and individual differences among phytoestrogen metabolites are important issues when researching the health effects of phytoestrogens in humans.

## 3. Phytoestrogen Mechanisms of Action—Anticancer Related Effects

In recent years, significant efforts have been made to elucidate the molecular mechanisms underlying the biological effects of phytoestrogens in both physiological and pathological conditions. The bioavailability and metabolism of phytoestrogens, as well as their effects on enzymes, nuclear receptors, and intracellular transduction mechanisms, play a crucial role in determining the overall impact of these compounds on cancer risk and progression [8]. However, the debate surrounding these effects persists, and further clarification is still needed. Phytoestrogens show a complex mode of action via interaction with the ER subtypes (i.e., ERα and ERβ), acting as either estrogen, triggering receptor pathways, or anti-estrogens, blocking normal estrogenic activity [10,40,41].

The dichotomy of ER modulating action induced by phytoestrogens led to the insertion of these compounds into the class of selective ER modulators (SERMs) [42,43,44,45] and probably provides an explanation regarding the conflicting evidence about the risks and benefits of these molecules on human health [46]. The activation of ER signaling pathways plays a vital role in the malignant progression of multiple cancers by comprehensively regulating downstream genes. The two ER subtypes have been described with different tissue distribution and ligand-binding affinities. ERα is mainly found in breast and uterine tissues and has been associated with pro-oncogenic responses while ERβ is the predominant isoform in the brain, bones, and blood vessels and is related to tumor-suppressive responses [19,47]. The alteration of the ERα/ERβ ratio in the affected tissues is one of the main reasons for the variability of estrogen-dependent cancer biology [48] and correlates with the response to the treatments and prognosis [49,50].

Phytoestrogens are known to bind ERs with much lower affinities than that of E2 (from 1/100 to 1/10,000), suggesting their weak estrogenic activities [19,47,51].

Moreover, unlike E2, which binds both ERα and ERβ with similar affinity, several phytoestrogens, including genistein, daidzein, and naringenin, display a substantially higher affinity for ERβ [19,52,53,54] (Table 1). Hence, considering the contrasting pro- and anti-cancerous effects exerted by ERα and Erβ, respectively, along with the unique expression patterns of these receptors in various cell types and tissues, the specific impact of phytoestrogens on each ER subtype becomes crucial in shaping the effects of these compounds on cancer progression [55].

The actions of phytoestrogens via ERs can be mediated by genomic and/or non-genomic mechanisms, in a dose- and tissue-specific manner [52,56]. The ER-mediated genomic effects of phytoestrogens result in the regulation of target genes, which include anti-inflammatory, anti-apoptotic, metabolic, and mitochondrial genes, as well as an improvement in mitochondrial biogenesis and function, which leads to increased resistance to stress [57,58] (Figure 2).

Phytoestrogens also modulate several therapeutically important oncogenic signaling pathways, including the epithelial–mesenchymal transition (EMT) and MAPK-associated pathways [46,59], and recruit transcription factors, such as response element binding protein (CREB), the activator protein 1 (AP-1), the stimulating protein 1 (Sp1), and the nuclear factor κ-light-chain-enhancer of activated B cells (NF-κB), which are correlated with cell cycle regulation, angiogenesis, metastasis, and apoptosis [60]. 

Besides nuclear events mediated by intracellular ER binding, many of these compounds also exert non-genomic effects through the activation of the membrane-associated ERs (mERs) and/or G-protein-coupled estrogen receptor 1 (GPER1/GPR30), which are involved in a diverse array of disorders, including cancer [61,62,63]. Among the membrane-initiated non-genomic effects is the activation of signaling cascades, such as the mitogen-activated protein extracellular kinase/extracellular signal-regulated kinase (MEK/ERK) and PI3K pathways, which affect cancer cell apoptosis and proliferation [10,64]. 

In addition, they can exert estrogenic activity by cross-talk with many other pathways, including those related to membrane-associated growth factor receptors, such as the human epidermal growth factor receptor (EGFR/HER) and the insulin-like growth factor 1-receptor (IGF1R) [65], as well as nuclear receptors, including [66] peroxisome proliferator-activated receptors (PPARs) [67] and estrogen-related receptor alpha/gamma (ERRα/γ) (Figure 2) [68]. Moreover, phytoestrogens can promote apoptosis and prevent the reproduction of malignant cells by blocking neo-angiogenesis, tyrosine-kinase, and topoisomerase proteins [69]. Several studies have also reported that, in addition to the classical estrogen receptor signaling and the genomic and non-genomic effects mentioned above, some phytoestrogens, including genistein and resveratrol, exert their anticancer effects by the epigenetic mechanism, such as the modulation of the chromatin structure [42,70] and the regulation of different cancer-associated miRNAs [71,72], suggesting new therapeutic strategies for cancer.

Phytoestrogens are mostly known for their potent antioxidant activity, i.e., another biological activity that indirectly reduces the risk of various degenerative diseases linked to oxidative stress, including cancer [73]. The chemical structures of these compounds consist of a long-conjugated system that includes phenolic groups. This structural arrangement confers them significant antioxidant properties [8], which have been linked to their chemoprevention potential, particularly in Asian populations. It is worth noting that in these populations, there is a correlation between soy consumption and a reduced occurrence of estrogen-related cancers [8,24].

On the other hand, at high concentrations, phytoestrogens may have pro-oxidant effects and induce cell death. Flavonoids autoxidize in aqueous medium and may form highly reactive radicals in the presence of transition metals. This effect has been described for several compounds, including genistein [74,75] and resveratrol [76], suggesting that a combination of phytoestrogens with anti-cancer treatments may render cancer cells more sensitive to treatment, in part by increasing the production of reactive oxygen species (ROS). However, given the high concentration of these compounds required for these activities, their impact on cancer onset and progression appears to be related to other cellular effects besides the modulation of oxidative stress [77,78].

Some phytoestrogens have also been shown to possess anti-inflammatory properties and modulate immune responses. They can inhibit the production of inflammatory mediators and reduce the expression of pro-inflammatory genes, contributing to their potential as anticancer agents [8,15].

As a whole, phytoestrogens exert a plethora of effects through multiple synergistic signaling pathways, which contribute to the outcome of phytoestrogen exposure on health and/or cancer cells. The specific effects of phytoestrogen exposure on cancer initiation, progression, and development may differ depending on the ERα/ERβ ratio in the affected tissue and the different selectivity and concentration of phytoestrogens [48]. In this regard, the majority of reported findings indicate distinct effects at low and high concentrations of phytoestrogens, which may be attributed to the capacity of these molecules to interact with and modulate ERs, thereby influencing endocrine functions [79]. Indeed, some studies have raised concerns about the potential adverse effects of soy products, particularly in high doses or when consumed by individuals with hormone-sensitive cancers [10,42]. Hence, it is crucial to gain a comprehensive understanding of how phytoestrogens interact with the ERs to fully evaluate their toxicologic and pharmacologic properties.

## 4. Molecular Basis of Osteosarcoma Pathogeneses

The difficulty in establishing an efficacious OS therapy is linked to the unclear specific markers for diagnosis and treatment. It is also due to the complexity of the OS genome, low incidence of this tumor, and significant biologic differences between OS subtypes. Nevertheless, the heterogeneity in the genotype of OS has translated into several expression profiles of macromolecular biomarkers, which are helpful in the clinic [2,80,81,82]. There are many genetic mutations observed in OS patients. The p53 and retinoblastoma (Rb) genes are well-known tumor-suppressor genes. Both germline and somatic mutations of the p53 and Rb genes have been proven to be involved in OS pathogenesis [82,83,84]. Inherited cancer predisposition syndromes, such as Li–Fraumeni, hereditary retinoblastoma, Rothmund–Thomson, Bloom, or Werner syndrome, may also influence the high appearance of this kind of tumor in young patients [83,85,86,87,88]. Among other genes mutated in more than 10% of OS cases, c-Myc plays a role in OS development and promotes cell invasion by activating MEK–ERK pathways. A high expression of c-Myc in OS tumors correlates with the formation of metastasis and poor prognosis [89].

Several studies have consistently demonstrated that OS cells have the capacity to develop and secrete a range of growth factors that exert autocrine and paracrine effects. Vascular endothelial growth factor (VEGF), transforming growth factor (TGF), IGF-I and IGF-II, and connective tissue growth factor (CTGF) are deregulated in OS, which leads to tumor progression and growth in target cells [82,90,91,92]. Parathyroid hormone-related peptide (PTHrP) and its receptor have also been implicated in OS progression and metastasis development, with PTHrP conferring OS chemoresistance by blocking signaling via p53 [93]. Epigenetic events have emerged as significant risk factors for OS, since the DNA methylation pattern of specific genes or gene regions and histone modifications may be involved in tumor development [94]. In addition, a variety of studies have found abnormally expressed levels of micro-RNAs (miRs), which have the potential to become prognostic biomarkers of OS. Overexpression of this molecule results in proliferation, migration, and invasion of tumor cells [68,95]. Among the miRNAs deregulated in osteosarcoma are miR-421, miR-16, miR-200b, and miR-101 [81,96,97].

OS is a highly metastatic tumor, and pulmonary metastases are the most common cause of death [82,98]. The ability of OS cells to metastasize has been found to be correlated with multiple processes and various cytophysiological changes, including changing the adhesion capabilities between cells and the extracellular matrix (ECM) and disrupting intercellular interactions [99,100]. Degradation of the ECM and components of the basement membrane caused by the concerted action of proteinases, such as matrix metalloproteinases (MMPs), cathepsins, and the plasminogen activator (PA), can play a critical role in OS invasion and metastasis [100]. Moreover, in metastatic forms of OS, some specific genetic changes have been observed, which include upregulation of the Wnt/β-catenin and Src pathways, the neurogenic locus notch homolog protein 1 and 2 (Notch1/Notch2) receptors [101,102] together with the downregulation of the Fas/Fas ligand pathway (a cell death pathway), which increases the metastatic potential of human OS [103,104].

In both primary bone cancer and bone metastases, the bone remodeling process creates a favorable environment for tumor establishment and progression. Osteoblasts and osteoclasts are the primary regulators of bone metabolism [105]. Specifically, osteoblasts secrete multiple components of ECM and MMPs in the OS niche, which are rich promoters of OS development. Moreover, osteoclasts play a pivotal role as bone-resorbing cells, and significant osteolysis exhibited in some OS cases can be directly attributed to the heightened activity of osteoclasts [100]. 

It has been demonstrated that OS is a condition characterized by deregulation in the signaling triad, i.e., the receptor activator of nuclear factor kB Ligand (RANKL), its receptors RANK, and osteoprotegerin (OPG) [106,107]. In its canonical function, RANKL, which is secreted by osteoblasts, induces bone destruction by mature osteoclasts. In response, osteoblasts secrete the OPG–RANKL decoy receptor and in this way inhibit osteoclast differentiation and resultant bone resorption [106,108]. The RANKL/OPG ratio in the blood is increased in high-grade OS, leading to the establishment of a vicious cycle between pathological bone remodeling and OS growth [108]. RANKL/RANK-signaling regulates OS cell migration and tissue-specific metastatic behavior in the lungs, but has no direct impact on OS-associated bone destruction and does not impact OS cell proliferation [106,109]. Thus, osteoclast pathways of differentiation, maturation, and activation constitute another compelling therapeutic target since the inhibition of bone resorption at the tumor–bone interface may lead to reduced local OS invasion [106]. 

Among the possible mechanisms that contribute to OS development in the bone microenvironment are alterations in the osteogenic pathway, which lead to the differentiation of mesenchymal stem cells (MSCs) into mature osteoblasts [81,110]. Defects in osteogenic differentiation or exposure to new non-native stimuli, such as pro-inflammatory cytokines and pro-tumor agents, may cause an imbalance between cell differentiation and proliferation, thus contributing to a malignant phenotype. OS cells share more characteristics with undifferentiated osteo-progenitors than with differentiated osteoblasts, including a high proliferative capacity and resistance to apoptosis. Indeed, osteogenic regulators associated with mature osteoblast phenotypes, such as CTGF, RUNX2, alkaline phosphatase (ALP), osteopontin (OPN), and osteocalcin (OCN), are very lowly expressed in both primary OS tumors and OS cell lines [111].

Although not well understood, some of the potential defects in the MSC differentiation cascade may include genetic and/or epigenetic changes in Wnt signaling, Rb, and p53. These alterations may lead to uncontrolled cell proliferation and disrupted differentiation, thus producing a tumorigenic phenotype [81,110]. Interestingly, treatments of human OS cells with therapeutic agents, such as peroxisome proliferator-activated receptor (PPAR) agonists [111], growth factors (e.g., PTHrP) [112], and SERMs [113], enable terminal differentiation and subsequent tumor inhibition. Hence, a better understanding of the relationship between defects in osteogenic differentiation and tumor development is of fundamental importance for the treatment of OS and promoting differentiation offers a potential for disease control.

## 5. Estrogen Receptors as a Potential Target for the Treatment of Osteosarcoma

A number of studies have established a correlation between the rapid bone growth experienced during puberty, when endogenous sex hormones, growth hormones, and IGF1 levels are at their highest, and OS development [114]. During the growth phase, there is greater bone turnover, so the possibility for defects to occur in the differentiation process and in the signaling pathways is amplified [1].

The estrogen/ERs axis is widely recognized for its essential role in bone development and growth from puberty through adulthood, acting via a variety of mechanisms and cell types [14,115,116]. Both ER isoforms, α and β, have been identified in bone, although their levels are lower than in reproductive tissues and can be affected by many parameters including the cell differentiation state [116,117,118]. ERα and ERβ have been observed to antagonize each other (activation vs. repression of transcription) in bone cells [116]. Therefore, ER isoform-specific activation could potentially lead to large differences in the patterns of gene expression induced by the ER–isoform homodimers [119,120]. In this regard, further work still needs to be completed to identify different genes that are regulated by estrogens and SERMs, including phytoestrogens, in each bone cell type during different stages of differentiation.

Several lines of evidence have hinted that the estrogens, which play prominent and well-described roles in osteoclast biology [14], might have a therapeutic differentiating effect on OS cells [121]. Accordingly, the loss of estrogens or the impaired functionality of the ERs appears to be involved in bone-forming cancers such as OS [14,122,123].

The expression of ERs is regulated by the methylation state in the promoter region. Their lack of expression, caused by epigenetic silencing (methylation) in the promoter region, is associated with the development of bone cancer and the metastasis of tumors in bone [95,124]. In line with this evidence, ERs have attracted widespread attention as promising targets for treating OS.

ERα is essential in osteogenesis and regulates cell growth in various tumors, including OS [68,125,126]. The activation of the ERs, especially ERα, triggers the downstream Wnt/beta-catenin signaling cascade, which promotes osteogenesis [127]. Moreover, targeting ERα-sensitive OS cells treated with methotrexate enhances the cytotoxic effects on OS when combined with DOX treatment [128].

OS typically originates from osteoblasts and/or MSCs, both of which normally express ERα [116,129]. However, the majority of OS tumors do not exhibit ERα expression, primarily due to the hypermethylation occurring in the ERα promoter region [122,124,125]. Notably, 143B cells demonstrate complete methylation of the ERα promoter, while U2OS and MG63 cells display partial methylation [122]. Interestingly, the use of demethylating agents to reverse the epigenetic silencing had a noteworthy effect on OS cell lines in vitro. It promoted differentiation and suppressed proliferation, and these effects were partially replicated in vivo, demonstrating potential disease control [122]. This suggests that the presence of ERα could serve as a therapeutic target and a prognostic factor for predicting the response to chemotherapy [122,125].

Regarding ERβ, while its specific functions have not been as extensively studied as those of ERα, evidence has emphasized its role as a tumor-inhibiting factor in estrogen-sensitive malignant tumors [130]. The anti-tumor effects of ERβ have also been reported in OS [13,113,131,132]. Specifically, it was found that ERβ mediates the proliferation, migration, and invasion of U2OS cells by regulating the integrin, IAP, NF-kB/Bcl-2, and PI3K/Akt signaling pathways. Moreover, silencing of Erβ promoted the metastatic phenotype in U2OS cells and OS tumors of mice, through the activation of the Wnt signaling pathway [133]. ERβ also plays an important role in inducing autophagy in U2OS cells by downregulating the expression of P62 and the phosphorylated mammalian target of rapamycin (p-mTOR) [13,134]. These findings suggest a role of ERβ agonists as potential candidates for therapy in OS. This is a noteworthy aspect considering that many phytoestrogens exhibit a higher binding affinity for ERβ compared to Erα [19]. 

The challenge in OS therapy lies in the lack of clearly defined markers for diagnosis and treatment. Because of the critical role of ERs in bone formation, whether or not the control of ERs by SERMs can modulate new bone formation and affect the prognosis or chemosensitivity of bone tumors is an interesting subject of study. 

## 6. Anti-Osteosarcoma Effects of Flavonoids

### 6.1. Genistein and Related Isoflavones

Genistein is a 15-carbon skeleton compound, chemically known as 5,7-dihydroxy-3-(4-hydroxyphenyl)chromen-4-one, which was first isolated from *Genista tinctoria*. It is the most prominent isoflavone from soy and soy-based food products [135,136]. 

In addition to genistein, soy foods are rich in daidzein, which differs from genistein in its lack of hydroxyl group at position 5 [137,138]. Genistein and daidzein derive from the O-methylated precursor biochanin A and formononetin, respectively [138,139] (Figure 3), which are generally less prevalent in soy and are found mostly in clover and alfalfa sprouts [138,140]. They are present as either aglycons or as glycosides [140,141]. The presence of hydroxyl groups and sugars increases their solubility in water, whilst methyl groups confer them lipophilicity [21]. 

Genistein and its related compounds are well known for their weak estrogen-like activity in mammals, including the prevention of bone loss and other estrogen decrease-related conditions [142]. Indeed, data have revealed a reduced occurrence of breast and prostate tumors as well as osteoporosis in human populations that consume a soybean-rich diet [143]. Isoflavones interfere with ERα and ERβ isoforms with higher binding affinity for the latter, particularly genistein (20- to 30-fold higher) and daidzein (five-fold) [19] (Table 1), producing distinct clinical effects from estrogens and contributing to the treatment of various cancers, such as gastric, breast, prostate, and non-small cell lung cancer prostate cancer [136,144,145,146].

Consistent with the above-reported ERβ anti-proliferative effects [10], it is believed that the chemotherapeutic potential of isoflavones is mainly due to their interactions with ERβ and vary in different cancer types, based upon their specific selectivity for target cells and their concentration, highlighting the complexity of dietary isoflavone functions [147,148,149]. ER-independent signaling mechanisms have also been described for isoflavones, including protein kinase regulation, enzymatic inhibition, growth factor modulation, antioxidant activity, or epigenetic changes [42].

Among the soybean isoflavones, genistein has shown to be one of the most potent inhibitors of cell proliferation in vitro by affecting protein tyrosine kinase (PTK) activity [150], suppressing the mammalian DNA topoisomerase II [136], and interacting with ERs on the nuclear envelope to promote G2/M phase arrest in various types of cancer cell lines [151,152]. The anticancer efficiency of genistein has been proven in numerous preclinical investigations [136]. Genistein from soy extracts, its free form, and its glycoside genistin are readily bioavailable and well tolerated with minimal toxicity [136,139]. Studies on OS cells suggest that genistein and its naturally occurring prenylated derivatives exert an inhibitory activity by binding ER isoforms with a statistically significant effect from 1 nmol/L [153]. The specific activation of ER signaling by genistein has been reported to be involved in the downregulation of the expression of the EGFR gene. This influences events strictly controlled by its signaling pathway, such as the differentiation of U2OS human OS cells that stably express ERs [154]. Indeed, previous studies have demonstrated that genistein and daidzein can induce osteoblast differentiation through the enhanced expression of ALP, bone morphogenetic protein 2 (BMP-2), and OPG, [155,156], which suggests that soybeans are able to prevent osteoporosis. In addition, the treatment of osteoblasts with genistein and other soybean isoflavones induces calcified bone noduli [157]. The osteogenic activity of genistein was also observed in the human MG-63 OS cell line [158] and the murine cell line LM8 [159] using a variety of approaches. Treatment with genistein switched the osteoblasts towards a more differentiated phenotype that was able to synthesize many of the essential factors required for the production of a new mineralized extracellular matrix, such as collagen type I (Col I) and ALP, even in a non-osteogenic growth medium [158].

As mentioned earlier, extensive vascularization and rapid growth are considered to be associated with the high metastatic potential and recurrence rate of OS [82,98].

The modulation of ECM components is a key element in cancer progression and invasion [99,100]. Glycosaminoglycans (GAGs)/proteoglycans (PGs) are the major macromolecules composing ECM [160]. Their localization can be both extracellular and cellular (cell membrane and intracellular granules), and they participate in the regulation of various cellular events, such as cell adhesion, migration, differentiation, and proliferation, critically affecting broader aspects in cancer initiation and progression [160]. In vitro studies have shown that the pathogenesis of OS implies qualitative and quantitative changes in PGs, which have important consequences on cell proliferation and/or differentiation [161]. Nikitovic and coauthors [162] investigated the activity of genistein on the synthesis and distribution of GAGs/PGs in the MG-63 and SaOS-2 cell lines, which differ in the density of the ERs they express. They observed a dose-dependent inhibitory effect of genistein on OS cell growth, which was accompanied by a reduction in both secreted and cell-associated GAGs/PGs by both cell lines, so suggesting a PTK mechanism. On the other hand, the MG-63 cell line showed a complex pattern of PG synthesis, indicating a dual action of genistein via the PTK mechanism and through the ER receptors, which are present in much higher density in MG-63 cells as compared to SaOS-2 cells [162].

Other than ERs, genistein can also regulate other nuclear malignancy-related receptors, such as peroxisome proliferator-activated receptor γ (PPARγ), an isotype of PPARs, which may further explain its activity against a range of cancers, including OS [163,164]. In OS, PPARγ plays a key role in suppressing cell proliferation and promoting osteoblastic terminal differentiation, suggesting a potential use of PPARγ agonists as chemotherapeutic and/or chemopreventive agents for human OS [111]. It has been reported that genistein treatment triggers growth inhibition and G2/M cell cycle arrest in human MG-63 cells by acting as a non-toxic activator of PPARγ expression [164]. In addition, the genistein-induced PPARγ signaling resulted in the downstream activation of the tumor suppressor phosphatase and the tensin homolog (PTEN), which in turn potently regulated the PI3K/Akt pathway involved in cell proliferation and survival [164]. Additionally, the anti-OS activity of genistein was examined in SaOS-2 and MG-63 OS cells in combination with calcitriol [165], the most active component of the vitamin D family [166]. The synergistic action of both of these substances had already proven effective in preventing osteoporosis and reducing hip fracture risk in postmenopausal women [167]. In OS cells, combined treatment significantly increased the expression of the enzyme sphingosine-1-phosphate lyase (SGPL1) [165], which irreversibly degrades sphingosine-1-phosphate (S1P) and whose production and secretion is associated with an increased capability of migration and invasion of cancer cells [151]. Besides the synergistic effects on the proliferation behavior, co-treatment increased the expression of the vitamin D receptor (VDR) and ERβ and influenced the cellular metabolism by decreasing extracellular acidification (which is a measure of glycolytic flux in cancer cells) as well as cell respiration rates (a measure of mitochondrial respiration) [165].

Gemcitabine is one of the commonly used anti-metabolite drugs. It is an analog of cytosine arabinoside which shows pronounced anti-tumor activity, in vitro and in vivo, in a variety of solid tumors, including OS [168]. Data reported by Zhang et al. showed that the combination of gemcitabine and genistein enhance the antitumor efficacy of gemcitabine in MG-63 and U2OS cell lines and helps to overcome resistance to gemcitabine treatment [169]. The induction of NF-kB activation and upregulation of its target genes by many anticancer agents, including gemcitabine, may mediate the cellular resistance to anticancer drugs [170]. In contrast, the activation of the Akt and NF-kB signaling pathways can be inhibited by genistein [144]. Combined treatment of genistein with gemcitabine reversed the cancer’s resistance to gemcitabine by abrogating the Akt/NF-κB pathway, which led to a significant reduction in cell viability and induction of apoptosis in OS cell lines [169,171]. The structure of genistein and some of its potential effects on OS cells are reported in Figure 3 and Table 2.

#### 6.1.1. Daidzein

The analog of genistein, daidzein (7,4-dihydroxyisoflavone) (Figure 3), is a well-studied isoflavone with a high nutritive value. It is predominantly found in soy and many unfermented foods not only in the form of daidzein, a glycoside conjugate, but also as acetylglycoside and aglycone [197]. The complex pharmacokinetic features of daidzein, along with its insolubility in water and oil, have blocked their use as a highly common compound in medicine or as a nutraceutical [197]. 

Although daidzein has been proven to have in vitro antitumor effects on a variety of cancers [8,198,199,200], there are few reports concerning its inhibitory activity towards OS [154,172] (Table 2). Daidzein inhibited proliferation and cell cycle progression and promoted apoptosis in the U2OS cell line that stably expresses either ERα or ERβ [154]. In addition, similarly to genistein, daidzein is able to modulate the expression level of EGFR, an estrogen-responsive gene, through the specific activation of both ER isoforms, thus affecting multiple intracellular events that are tightly controlled by the EGFR transduction pathway, such as the maturation of osteoblasts. 

The link between the ER and EGFR pathways was confirmed by treatment with 4-hydroxytamoxifen (4OH-T), a synthetic estrogen receptor ligand, which regulated the EGFR level producing differentiation and proliferation effects via ERs. Since the treatments did not induce any significant change in ER-negative U2OS cells, the results found in the U2OS-ERα and U2OS-ERβ cell lines could be totally ascribed to ER-mediated effects [154]. Recently, Zhu and coworkers investigated the underlying mechanism of daidzein against OS by means of an in-depth systematic pharmacological analysis, which identified the Src-MAPK pathway as the highest-ranked target of daidzein in 143B and U2OS cells and in in vivo OS models [172]. 

#### 6.1.2. Biochanin A

Biochanin A (5,7-dihydroxy-4′-methoxy-isoflavone) (Figure 3) is a phytoestrogenic isoflavone found in legumes, particularly red clover (*Trifolium pratense*), but it is also present in cabbage, alfalfa, and many other herbal products [152]. The biological effects of biochanin A observed in vitro and in vivo are different from those observed for its derivative genistein. It plays complex roles in the regulation of multiple biological functions by binding DNA and some specific proteins or acting as a competitive substrate for some enzymes. Biochanin A extract from plants is already commercially available because of its potential benefits to human health despite its limited bioavailability [201,202]. It is used for the reduction in oxidative stress, treatment of osteoporosis, anti-inflammatory effects, regulation of blood glucose levels, and for treatment of allergies [203,204]. Moreover, like other isoflavones, biochanin A exhibits chemopreventive activity against various cancers [202]. 

Several studies have suggested that biochanin A has the potential to prevent and treat OS [173,174] (Table 2). The antiproliferative potential of biochanin A and its underlying mechanisms in human OS have been investigated by Hsu and et al. They reported that biochanin A dose-dependently inhibits cell growth and induces apoptosis in the MG-63 and U2OS OS cells by triggering the activation of the intrinsic mitochondrial pathway and caspase-9 and -3 and increasing the Bax: Bcl-2/Bcl-XL ratio. 

DOX is probably the most studied drug in combination with phytoestrogens. It is an antitumor antibiotic which acts by interfering with the enzymes involved in DNA replication or by causing strand breakage [205]. Hsu et al. also demonstrated that a combination of the chemotherapeutic agent DOX with biochanin A had synergistic cytotoxicity [173]. Supporting evidence for the anti-OS activity of biochanin A has been recently reported by Zhao et al. [174]. Biochanin A treatment clearly increased apoptotic rates and decreased migration and invasion abilities in MG-63 and U2OS cells, in a time- and dose- dependent manner. Relevant genes involved in cell proliferation, apoptosis, invasion, and migration (i.e., proliferating cell nuclear antigen (PCNA), caspase-3, cyclinD1, Bcl-2, MMP-9, N-cadherin, and E-cadherin) showed altered expressions in both OS cell lines [174]. E-cadherin is an important factor of cell–cell adhesion. It is classified as a cancer depressor because the loss of E-cadherin may lead to the destruction of the cytoskeleton, promoting cell invasion and migration [206]. The expression of N-cadherin in epithelial tumors characterizes tumorigenesis as it may induce EMT, reinforce tumor cell activity, and promote the interaction between tumors and neighboring cells [207]. OS cells treated with biochanin A exhibited a decreased expression of MMP-9 and N-cadherin, while showing increased expression of E-cadherin. These findings strongly indicated that biochanin A may possess a suppressive function in cancer invasion and migration by mediating the process of tumor EMT [174].

#### 6.1.3. Formononetin

Formononetin (7-hydroxy, 4′-methoxy isoflavone) (Figure 3) is the active ingredient of the traditional Chinese medicines *astragalus*, *angelica*, and *Pueraria lobate*. Formononetin exhibits anticancer effects against ovarian cancer, colorectal cancer, and gastric cancer by suppressing cell viability and inducing apoptosis through the regulation of the estrogen-dependent signaling pathway [208,209].

Reports have also documented the anti-OS activity of formononetin (Table 2). It promoted the apoptosis of human bone cancer in vitro and in vivo in nude mice that had undergone orthotopic tumor implants by modulating the expression levels of the apoptosis-related factors ERK, Akt [176], Bcl-2, Bax, and caspase-3 and decreasing the level of miR-375, an ER signaling-related miRNA, in ER-positive U2OS cells [175]. Further evidence showed that in MG-63 cells, the anti-proliferative function of formononetin is related to the upregulation of the expression of the tumor suppressor PTEN gene, via miR-214-3p [177], which is one of the miRs with oncogene properties that is considerably increased in OS [210]. Recently, bioinformatic-based network pharmacology has been used to disclose other therapeutic targets and bio-mechanisms of anti-OS formononetin activity [178]. The biological processes of formononetin against OS were principally linked to the regulation of cell motility, cell proliferation, and the regulation of gene expression. The main core targets of formononetin were determined as estrogen receptor 1 (ESR1), TP53, and receptor tyrosine-protein kinase erbB-2 (ERBB2). Interestingly, they were representatively validated following formononetin treatment in vivo in tumor-bearing mice and clinical cases, suggesting that these predictive targets might be potential biomarkers for the screening and treatment of OS [178].

### 6.2. Flavonols

#### 6.2.1. Quercetin

An intake of flavonols is found to be associated with a wide range of health benefits, which include antioxidant potential and anti-inflammatory effects. Flavonols may also play a role in reducing the risk of chronic diseases, such as cardiovascular disease, cancer, and neurodegenerative disorders [211].

Quercetin (3,5,7,3′,4′-pentahydroxyflavone) (Figure 4) is the most abundant plant pigment in the extensive class of flavonols. It can be found in numerous plant species and in daily foods, such as vegetables (capers contain the greatest quantity in relation to their weight), fruits, nuts, and teas. The average daily dietary intake of quercetin is estimated to be 16 mg [212]. Due to its potent anti-inflammatory and antioxidant effects, quercetin is considered an advantageous agent for therapeutic purposes for a number of diseases, including cardiovascular diseases, arthritis, allergies, and diabetes [213]. Indeed, it is commercially accessible as a supplementary agent. Oral administration of 1 g of quercetin per day is safe and is absorbed up to 60% [214].

Extensive studies have demonstrated the efficacy of quercetin for the prevention and treatment of several types of cancer, in in vivo and in vitro models [215]. Quercetin can significantly prevent the cell cycle, promote cell apoptosis, and suppress tumor invasive behavior via a variety of mechanisms [216]. There is evidence that several of the effects of quercetin on the survival of cancer cells might rely on ER-dependent mechanisms. 

In particular, it has been shown that quercetin can impair the ERα-mediated rapid signaling, preventing the anti-apoptotic and proliferative ERK/MAPK and PI3K/Akt pathway activation and sustaining the persistent phosphorylation of p38/MAPK and, in turn, the blockage of cell cycle progression and the induction of the pro-apoptotic cascade, without affecting the transcriptional effect of activated ERα [217]. Thus, quercetin can influence cancer cell proliferation and survival by acting as partial antagonists of ERα-activated rapid signals [218,219]. To the contrary, quercetin can behave as an E2-mimetic agent in the presence of ERβ by activating the p38/MAPK and the downstream pro-apoptotic caspase-3 activation and poly (ADP-ribose) polymerase (PARP) cleavage. This indicates the pivotal role of both ER subtypes and a differential agonist or partial antagonistic effect of quercetin in the definition of its anti-carcinogenic potential [220].

The strong cytotoxic activity of quercetin against OS has been largely demonstrated (Table 2). Although the related studies are mainly limited to in vivo and in vitro investigations, findings are promising. Quercetin treatment results in the suppression of proliferation, cell cycle arrest, induction of apoptosis, and reduced potential for adhesion and migration in several human OS cells lines, including 143B [182,186], HOS, MG-63 [179,184], U2OS, and SaOS-2 cells [185]. Regarding the effect on OS cell growth, a number of key elements of the cellular apoptotic signaling pathway seems to be involved in the quercetin-dependent regulation of apoptosis. In an early study, Liang and coworkers [179] evaluated the effects of quercetin on the viability of human OS MG-63 cells. Quercetin treatment resulted in a decreased expression of the anti-apoptotic protein Bcl-2, which was paralleled with the increase in the pro-apoptotic proteins BAX and cytochrome C, activation of caspase-3 and -9, and loss of mitochondrial membrane potential, indicating that quercetin was able to promote apoptosis via activation of the mitochondrial-dependent pathway [179]. Additional evidence demonstrated that the administration of quercetin to HOS and ATCC 1543 human OS cell lines leads to cell cycle arrest at the G(1)/S phase accompanied by the downregulation of cyclin D1, one of the cyclins required for G(1) to S progression. Subsequent apoptosis was induced by gradual activation of caspase-3 and subsequent PARP cleavage [180].

The ability of quercetin to induce apoptotic cell death has also been associated with overcoming drug-resistance in OS cells. High-dose methotrexate (HDMTX) is considered to be a key agent in determining the chemotherapeutic outcome of OS patients [221]. However, drug resistance often develops in the late stage of treatment. Data reported by Xie et al. [181] revealed an apoptosis-inducing activity of quercetin in an MTX-resistant OS model (U2OS/MTX300 cells). Exploring the mechanism underlying these effects, the authors found that cell death was accompanied by mitochondrial dysfunction and dephosphorylation of Akt, suggesting that quercetin-induced apoptosis might be associated with the apoptosis pathway of mitochondria and Akt activity [181]. Similar evidence was provided by Yin et al. [222], further sustaining the role of quercetin as a potential chemotherapeutic agent for MTX-resistant osteosarcoma. 

OS is known to be a disease with a high propensity for metastasis to the lung [5]. In comparison to other cell lines, only 143B cells are able to generate a reliable and reproducible in vivo mouse model that develops primary tumors after intratibial injection within three to five weeks [223]. This mouse model also more closely reflects human disease because it develops metastases in the lung, which is the main location of metastasis in humans [224]. An early study by Berndt et al. [182] evaluated the anticancer properties of quercetin in human a 143B OS cell line. Quercetin treatment caused an arrest of 143B cells in the G2/M transition of the cell cycle. The cell cycle arrest was followed by cell death via activation of the apoptotic signaling pathway, as shown by a dose-dependent caspase-3, caspase-7, and PARP cleavage after 36 h of incubation with quercetin. The pan-caspase inhibitor Z-VAD prevented PARP cleavage-dependent apoptosis. Quercetin also effectively blocked some hallmarks of metastatic behavior, such as adhesion and migration. Overall, these findings suggested a role for quercetin as a potential drug that can target cells of the primary tumor and metastasizing foci [182].

Quercetin’s ability to induce cell death in OS cell lines is not limited to apoptosis. A recent study by Wu et al. [183] showed for the first time a role for quercetin in promoting autophagic cell death in human OS cells. It is well accepted that autophagy plays a two-faced role in cancer as a tumor suppressor or as a pro-oncogenic mechanism. Indeed, there is a dynamic relationship between the rate of protein degradation through autophagy, i.e., autophagic flux, and the susceptibility of tumors to undergo apoptosis which is critical in the autophagy/cancer relationship [225]. Therefore, an accurate modulation of this pathway is the challenge faced when targeting autophagy in the clinical setting [226]. 

The quercetin dependent increase of autophagic flux was shown by a dose-dependent upregulation of LC3B-II/LC3B-I ratio, a hallmark of autophagy, in quercetin-treated MG-63 cells. Moreover, pharmacological inhibition of autophagy or genetic blocking autophagy by autophagy-related gene 5 (ATG5) knockdown efficiently protected against quercetin-induced cell death, supporting the role of quercetin in triggering autophagic cell death in MG-63 cells [183]. The authors also investigated the effect of quercetin exposure on transcription factor nuclear protein 1 (NUPR1) activity [183]. NUPR1 is a master regulator of the autophagy flux typically expressed in response to stress signals induced by genotoxic signals and agents [227]. Under stressful conditions, such as chemotherapy treatment, an interplay between the homeostatic NUPR1 and autophagy pathways may occur in cancer cells, ultimately dictating their fate between cell death or survival [227]. Data revealed that quercetin increased NUPR1 expression and activated NUPR1 reporter activity, which contributed to the expression of autophagy-related genes by disturbing ROS homeostasis, indicating an important role for NUPR1 signaling in quercetin-induced autophagy in MG-63 cells [183].

Activating local cell invasion and distant metastasis represents another important hallmark of cancer that mainly reflects the progression of carcinomas to a higher grade of malignancy. Cell migration is a critical process for cancer cell spread, invasion, and distant metastasis [228]. Hypoxia-inducible factor (HIF)-1α was shown to be correlated with tumor grade, metastasis, and poor outcomes in various cancers including OS. 

Its upregulation results in increased cell proliferation and migration, as well as the development of chemotherapeutic resistance [229]. Lan et al., [184] reported a dose- and time-dependent reduction in cell migration and invasion in human HOS and MG-63 cell lines, which was paralleled with a significant quercetin-induced downregulation of the expression of HIF-1α and downstream genes, such as, VEGF, MMP-2, and MMP-9, which play essential roles in promoting cancer invasion and metastasis. In addition, quercetin treatment suppressed the formation and proliferation of metastatic lung tumors in vivo in an OS nude mouse model [184]. In U2OS and SaOS-2 cells, the quercetin-mediated inhibition of the metastatic phenotype was associated with a significant downregulation of parathyroid hormone receptor 1 (PTHR1) [185], an important G-protein coupled receptor involved in OS pathophysiology [230]. 

Many lines of compelling data indicated the role of quercetin in sensitizing cancer cells to the action of several anti-cancer drugs [231,232]. In this respect, synergistic anti-tumor activities of quercetin and CDDP, a widely used chemotherapeutic agent, have been reported in cancer treatment in several in vivo and in vitro models [231,232,233]. 

Regarding OS, the efficacy of quercetin in enhancing CDDP sensitivity has been demonstrated by Zhang et al. in 143B cells [186]. miR-217 is a tumor suppressor which inhibits cell proliferation and metastasis in OS. Its over-expression could reverse CDDP chemoresistance in lung cancer cells [234]. Expression of miR-217 was upregulated after quercetin and/or CDDP treatment, while its target, Kirsten rat sarcoma virus (KRAS), was downregulated both at mRNA and protein levels, thus suggesting an involvement of the miR-217-KRAS axis in the QUE-improved sensitivity of CDDP [186]. Quercetin has also been shown to be effective in boosting the anticancer activity of MTX on SaOS-2 cancer cells. A decline in MTX IC50 value was observed from 13.7 ng/mL to 8.45 ng/mL in the presence of quercetin. Moreover, the mRNA expression outcomes indicated that the combination therapy significantly upregulates the tumor suppressor genes, such as p53, CBX7, and CYLD, and declines anti-apoptotic genes BCL-2 and miR-223, which can lead to proliferation, inhibition, and apoptosis inducement [187].

Taken together, these findings provide experimental evidence of the effectiveness of quercetin in treating OS in human cell lines. On the other hand, further research, including clinical trials, is needed to support the future development of quercetin as an effective and safe candidate agent for the prevention and/or therapy of OS. 

#### 6.2.2. Galangin

Galangin (3,5,7-trihydroxyflavone) (Figure 4) is another plant flavonol from the flavonoid group of polyphenols. It is primarily extracted from the rhizome of *Alpinia officinarum*, which has been used as an herbal medicine in Asia for decades. Studies have shown that galangin has anti-inflammatory [235], antibacterial [236], and antiviral [237] activities in vitro. Currently, its antitumor properties are the subject of attention. Studies have shown that galangin suppresses the proliferation and functions of various tumor cells [238,239,240]. However, research on galangin’s effects on OS, specifically, is limited (Table 2). In a study conducted by Yang et al. [188], exposure to galangin significantly suppressed MG-63 and U2OS cells by inhibiting their proliferation and invasion and triggering their apoptosis in a concentration-dependent manner. The underlying mechanism was associated with the suppression of PI3K and its downstream regulators, cyclin D1 and MMP-2/9, and upregulation of p27Kip1, caspase-3, and caspase-8 [188]. Other evidence indicated that, in addition to effectively attenuating cell proliferation, incubation of MG-63 and U2OS cells with galangin increased dose-dependently the expression levels of several markers for osteogenic differentiation, such as Col I, ALP, OPN, OCN, and the transcription factor Runx2. Galangin could also attenuate OS growth in vivo, in the xenograft mouse model [189]. There are several cytokines that control bone formation, among which TGF-β1 has been proven to be fundamental in osteoblastic differentiation and bone matrix synthesis, through Smads-dependent signaling [241]. TGF-β1 secretion and the phosphorylation of Smad2 and Smad3 were triggered in a dose-dependent manner after galangin treatment, clearly indicating that galangin-mediated cell differentiation was dependent on its selective activation of the TGF-β1/Smad2/3 signaling pathway [189].

### 6.3. Apigenin

Apigenin (4′,5,7-trihydroxyflavone) is the main compound in the flavone group (Figure 4). It is widely contained in many fruits and vegetables, such as oranges, tea, chamomile, onions, and wheat sprouts [242]. Apigenin is thought to protect cells against oxidative damage by enhancing mitochondrial function [243]. Furthermore, it possess anticancer properties in vitro and in vivo, inducing cell cycle arrest and DNA damage in different types of cancer cells [242,244,245].

Results reported by Lin et al. showed that apigenin triggers apoptosis in U2OS cells and inhibits xenograft tumor growth [190]. Liu et al. [191] confirmed the cytotoxic role of apigenin on both U2OS and MG-63 cells and investigated the underlying molecular mechanisms of its anti-OS effect. 

The canonical Wnt-β-catenin signaling pathway is widely expressed in bone tissue and cells, and its deregulation is closely associated with the progression of OS [101,102,246]. It has been reported that a decreased β-catenin expression can downregulate MMP-14 expression, thereby resulting in suppression of the invasion and motility of MG-63 cells [247]. Apigenin was able to decrease the expression of β-catenin. Moreover, overexpression of β-catenin reversed the inhibitory effect of apigenin on OS cells and knockdown of β-catenin-enhanced apigenin-inhibited proliferation and invasion in OS cells. These results supported the hypothesis that the Wnt/β-catenin pathway is involved in OS cell proliferation and invasion in response to apigenin [191].

### 6.4. Naringenin

Citrus fruits (grapefruit and oranges) and tomatoes are rich sources of flavanones, which are known for their numerous health benefits due to their ability to scavenge free radicals in various basal metabolic conditions [24]. 

Naringenin (4′,5,7-trihydroxyflavonone), especially abundant in the Mediterranean diet, is the most extensively studied flavanone (Figure 4). Despite its limited bioavailability [248], it shows great promise in various therapeutic applications. Its notable benefits in in vivo and in vitro models, which have been comprehensively reviewed by Arafah et al. [249], include anti-inflammatory, antioxidant, and anticancer properties. However, a scarce number of clinical studies have been conducted to date compromising its commercial exploitation [248]. The generation of ROS is one of the key mechanisms responsible for promoting different stages of cancer [250]. Naringenin, as an antioxidant, does efficiently counter such effects [249]. In this regard, it has been reported that the long-term administration of naringenin (20 mg daily) inhibits OS progression and local recurrence in the patients (*n* = 47) who underwent OS surgery by improving the antioxidant and anti-inflammatory capacities of OS patients [251]. 

The cytotoxicity of naringenin has been demonstrated in numerous in vitro studies using various kinds of cancer cells, e.g., breast, colon, and liver cancer cell lines [252,253]. Antioxidant activities and kinase and glucose uptake inhibition, have been proposed as molecular mechanisms for these effects. In addition, naringenin stimulation is believed to inhibit unregulated growth and induces apoptotic cascade in different cancer cell types by ERα or ERβ signaling [253]. Notably, naringenin shows an anti-estrogenic effect only in ERα containing cells, whereas in ERβ containing cells, naringenin mimics E2 effects [253]. A recent study conducted by Lee et al. [192] has shown for the first time the cytotoxic and antiproliferative effects of naringenin on human OS cells. Although a high naringenin concentration was used during the study, it selectively inhibited the growth of OS cells (IC50 values for HOS and U2OS cells were 276 and 389 μM, respectively) with less cytotoxicity in normal human bone cells. The suppression of cell growth was accompanied by a significant increase in intracellular ROS generation and mitochondrial dysfunction, resulting in the activation of endoplasmic reticulum stress-mediated autophagy and apoptosis. This evidence suggested that this flavanone exerts its mechanism of action in cancer OS cells through ROS-mediated endoplasmic reticulum stress signaling pathways [192]. 

### 6.5. Catechins

Catechins, which are considered readily applicable and safe phytochemicals [250], are the major bioactive constituent in green tea polyphenols. Their basic structure consists of a flavan-3-ol unit with a catechol (1,2-dihydroxybenzene) moiety. There are different types of green tea catechins (GTC), including epicatechin (EC), epicatechin gallate (ECG), epigallocatechin (EGC), and epigallocatechin gallate (EGCG), among others. These variations occur due to the number and position of hydroxyl groups attached to the flavan-3-ol structure. The arrangement of hydroxyl groups determines the distinct properties and potential health benefits of each catechin [251]. EGCG is the most abundant and biologically active catechin [252]. Its chemical structure is reported in Figure 4. Based on decades of research, EGCG has received considerable attention due to its inhibitory activities against carcinogenesis at all stages, i.e., initiation, promotion, and progression [253,254]. Other catechins, such as ECG and EGC, have been shown to have similar, albeit lower, activities in numerous studies [255,256]. However, in humans, plasma bioavailability of GTCs is very low, which has been in part attributed to their oxidation, metabolism, and efflux [257]. The low bioavailability and absorption of GTCs are considered to be the major reason behind the differing effects between in vitro and in vivo studies. In fact, extensive studies on the improvement of GTC bioavailability should help in this regard [257].

Bone strength mainly relies on selenium (Se), calcium, and vitamin (K and D) contents. Se deficiency is associated with the risk of developing multiple cancers, including OS [258]. To minimize the risks associated with Se deficiency, its doping with hydroxyapatite (HAp) can be an effective approach which may potentially reduce the growth of OS cells. Currently, HAp has received considerable attention in reconstructive surgeries, orthodontic, and three-dimensional printing of scaffolds, owing to its high bioactive and osteoconductive properties [259,260]. Given the known antitumor properties of catechins, a recent study by Khan et al. aimed to develop catechin-modified Se-doped HAp nanocomposites (CC/Se-HAp) for potential application in OS therapy [261]. Cell toxicity analysis showed that CC/Se-HAp were rapidly internalized into the MNNG/HOS cells and improved anticancer activity as compared to a Se-doped HAp nanocomposite, by inducing ROS-mediated apoptosis through the activation of the caspase-3 pathway [261]. This suggests that CC/Se-HAp nanoparticles possess the potential for the targeted treatment of OS. Further studies on the anti-OS properties of catechins have mainly focused on EGCG (Table 2). This green tea polyphenol can effectively inhibit the tumor characteristics of OS cells (i.e., 143B, MG-63, U2OS and SaOS-2), including proliferation, migration, and apoptosis [262,263,264]. Moreover, it protects local bone tissue from destruction and prevents lung metastasis of tumor cells [263,264]. These effects may occur by regulating the activity of the Wnt/β-catenin pathway, as demonstrated by cell and animal experiments [264]. Additional EGCG targets have been identified. In MG-63 and U2OS, the inhibitory activity of EGCG treatment has been partially associated with the upregulation of miR-1, one of the miRNAs critically involved in the pathogenesis and progression of human OS [263]. Moreover, the combined administration of EGCG and the IL-1 receptor antagonist (IL-1Ra) efficiently downregulated IL-1-induced IL-6 and IL-8 release from U2OS cells. This treatment approach also reduced the secretion of invasiveness-promoting MMP-2 and pro-angiogenic VEGF, without affecting the metabolic response and caspase-3 activity [262]. EGCG has also been shown to act as a chemosensitizer for DOX in OS models. The synergistic interaction between EGCG and DOX has been shown to be mediated by the SOX2 overlapping transcript (SOX2OT), which contributes to suppress OS via autophagy and stemness inhibition [265] (p. 7).

## 7. Anti-Osteosarcoma Effects of Non-Flavonoids

### 7.1. Stilbenes

#### 7.1.1. Resveratrol

Resveratrol (3,5,4′-trihydroxy-trans-stilbene) is a powerful nutritional polyphenol found in more than 70 plant species and is especially abundant in food products, such as red grapes (up to 14 mg/L) and their derived products, peanuts, mulberries, and soy [33,266]. Resveratrol is a phytoalexin. These chemicals are characterized by their low molecular weight and their ability to inhibit the progress of infections and other adverse stressful conditions for the plants [267]. Resveratrol is characterized by a stilbene structure which consists of two phenolic rings bonded together by a double styrene bond, which is responsible for the isometric *cis*- and *trans*-forms of resveratrol (Figure 5). The *trans*-isomer appears to be the more predominant and stable natural form [268]. There are many synthetic and natural analogues of resveratrol as well as adducts, derivatives, and conjugates, including glucosides [269]. In red and white grape juice, resveratrol exists mostly as polydatin, its glycosidic form (see next section). Red grape juices contain significant amounts of trans-polydatin, followed by cis-polydatin and trans-resveratrol. These compounds are considered the primary compounds responsible for the health benefits associated with wine consumption [266]. 

Resveratrol is commonly used as a nutraceutical in the management of high cholesterol, cancer, heart disease, and many other pathological conditions [270,271,272]. Its chemopreventive and anticancer effects have been documented in in vivo and clinical studies in a wide variety of tumor cell types, highlighting its role in diverse cellular events associated with all stages of carcinogenesis, i.e., tumor initiation, promotion, and progression [8,273,274,275,276]. Unlike other phytoestrogens which bind ERβ with higher affinity, resveratrol binds ERα and ERβ with comparable affinity but with 7000-fold lower affinity than estradiol, acting as mixed agonist/antagonist [52]. Many lines of compelling data indicate that the effects of resveratrol on the survival of estrogen-related cancer cells might rely on ER-dependent mechanisms [277,278].

Regarding the bone microenvironment, resveratrol has been shown to have multiple bioactivities, including antioxidative, anti-inflammatory, estrogen-like, and proliferative properties that can influence bone metabolism [105,279]. In particular, in normal osteoblasts and osteoclasts, it regulates cell proliferation, cellular senescence and apoptosis, and inflammation processes, reducing the activity of NF-B and MAPK, and also acts through an epigenetic control, modulating the expression and activity of sirtuin-1 (SIRT-1), which is capable of increasing osteoblast survival and differentiation [280]. 

A body of studies on the effect of resveratrol on OS demonstrated a strong suppression of cell viability as well as self-renewal ability and tumorigenesis of OS cells, whereas no significant inhibition effect on normal osteoblasts was observed [281,282,283,284,285,286]. However, the underlying mechanisms of action of resveratrol on OS cells have only been partially defined (Table 3). The Janus kinase 2/signal transducer and activator of transcription 3 (JAK2/STAT3) pathway is involved in different biological processes, such as immunity, cell division, cell death, and tumor formation [287]. Aberrantly activated JAK2/STAT3 is frequently detected in many cancer diseases that are usually refractory to standard chemotherapy [287]. Studies have demonstrated a critical role of STAT3 signaling in the persistence of cancer stem cells (CSCs) [288], which is a primary cause of tumor relapse and metastasis [289]. Peng et al. [284] investigated the underlying molecular mechanism of resveratrol activity against OS CSCs, reporting that resveratrol treatment inhibited tumorigenesis ability by decreasing the synthesis of cytokines, such as IL-6, IFN-γ, TNF-α, and Oncostatin M and preventing the JAK2/STAT3 signaling pathway, which was consistent with the decline of the CSC marker CD133. On the other hand, exogenous STAT3 activation had the opposite effect and could abrogate the effect of resveratrol on CSCs, suggesting that resveratrol may be a promising therapeutic agent for OS management [284]. 

Additional studies have reported a correlation between resveratrol activity and the canonical Wnt/β-catenin cascade [283,290]. Aberrant activation of the Wnt-β-catenin signaling pathway plays a critical role in OS pathogenesis, which makes the Wnt/β-catenin pathway a hot topic in OS research [101,102,246]. Nevertheless, the development of therapeutic agents specifically targeting the aberrant Wnt activation in OS cells is still in its infancy. Exploring the effects of the resveratrol treatment against U2OS cells, Xie and coworkers found that the antitumor effects of resveratrol occurred by suppressing the activity of the Wnt/β-catenin pathway and the expression of related genes, such as c-myc, cyclin D1, MMP-2, and MMP-9 [290].

Moreover, resveratrol was able to upregulate the expression level of Connexin 43 (Cx43) and E-cadherin [290], two important mediators of cell–cell adhesion which are critical in tumor progression [206,291]. The association between the anti-OS effect of resveratrol and the Wnt signaling was further demonstrated by Zou et al. [283]. 

Using β-catenin as a drug target, they performed a high content screening of botanical extracts to identify potential drugs against the human MG-63 OS cell line. β-catenin is a pivotal member of the canonical Wnt signaling pathway with the dual functions of regulating the coordination of cell–cell adhesion and gene transcription [292]. Aberrant expression of β-catenin activates numerous downstream target genes of the Wnt signaling pathway, a number of which are associated with cancer progression [293,294]. In a total of 14 botanical extracts assessed, resveratrol markedly downregulated the expression of β-catenin and significantly inhibited MG-63 cell proliferation [283].

OS is characterized by a high metastatic potential which is associated with a high death rate [82,98]. Proteinase enzymes, such as cathepsins, MMPs, and PA, are involved in many steps of tumor metastasis, including tumor invasion, migration, host immune escape, angiogenesis, and tumor growth [295]. MMPs, especially MMP-2 and MMP-9, are usually over-expressed in a wide range of human cancer types, including OS, providing a potential therapeutic target [100]. Based on in vitro, in vivo, and clinical evidence, Yang and coauthors showed that resveratrol can suppress the metastatic potential of human OS cells (i.e., HOS, MG-63, U2OS, Saos-2, and 143B) through transcriptional and epigenetic regulation of MMP-2, by, respectively, inhibiting cAMP CREB-DNA-binding activity and upregulating miR-328, which was initiated by the inhibition of the p38 MAPK/JNK pathways. Consistently, suppression of miR-328 significantly relieved MMP-2 and motility inhibition imposed by resveratrol treatment [296]. The proliferation, invasion, and metastasis of tumor cells, as well as tumor relapse, are strongly correlated with the interactions of several factors, in which angiogenesis is a prerequisite. During this process, VEGF functions as the most significant vascular endothelial stimulating factor [297,298]. According to Liu et al. [299], resveratrol exerts a time and dose-dependent inhibition of OS cell invasion capabilities and proliferation, which is mediated by the downregulation of VEGF expression [299]. 

Recently, De Luca and colleagues [286] made a series of important observations on different human OS cell lines (i.e., MG-63, SaOS-2, KHOS, U2OS). They found that resveratrol was involved in the *p*AKT and caspase-3 pathway, causing cell growth inhibition and increase in apoptosis. Moreover, a significant increase in osteoblastic differentiation genes, such as osterix (Osx), OPN, ALP, Col I alpha 1, and OCN, was observed, suggesting that resveratrol may act as an inducer of differentiation, which is known to make OS cells more vulnerable to the action of chemotherapeutic agents [300]. Furthermore, they highlighted an epigenetic action of resveratrol on the promoters of interleukins IL-6 and IL-8, whose role in tumor progression is a well-described process in several cancer models [301,302]. The epigenetic change induced the reduction in the secretion of interleukins IL-6 and IL-8, which further explained the inhibitory effects of resveratrol on OS cellular growth and motility. In line with data previously reported [281], the OS cell lines examined, which represent the various typical characteristics of OS, responded in a variable manner to resveratrol treatment. This explained why, by calculating the IC50 of resveratrol, different values were obtained, i.e., around 120 µM for MG-63 and SaOS-2 and around 60 µM for KHOS and U2OS treatment [286].

**Table 3 ijms-24-13344-t003:** Effects of non-flavonoids on osteosarcoma. Downward arrows represent downregulation or reduction. Upward arrows represent upregulation or increase.

Phytoestrogen	Cell line/ In Vivo Model	Concentrations	Combined Treatment	Molecular Mechanism	Observed Effects	References
**Resveratrol**	MNNG/HOS, MG-63 tumor xenograft mouse	10–40 μM 100 mg/kg/d		↑ caspase-3, Bax, cleaved PARP, ↓ Bcl-2, Bcl-xL; ↓ cytokines ↓ JAK2/STAT3 pathway	↑ apoptosis ↓ CSCs survival ↓ tumor growth	[284]
	U2OS	6–24 μg/mL		↓ Wnt/β-catenin pathway ↓ β-catenin, c-myc, cyclin D1, ↓ MMP-2/9 ↑ Cx43, E-cadherin	↓ proliferation ↑ apoptosis ↓ migration, invasion	[290]
	MG-63	10–40 µg/mL		↓ β-catenin signaling	↓ proliferation	[283]
	HOS, MG-63, U2OS, SaOS-2, 143B HOS orthotopic graft model	25–200 μM 40, 100 mg/kg/d		↓ p38 MAPK/JNK pathways ↓ CREB, ↓ MMP-2, ↑ miR-328	↓ migration, invasion, adhesion ↓ tumor growth, metastasis	[296]
	U2OS	10–40 μM		↓ VEGF		[299]
	MG-63, SaOS-2, KHOS, U2OS	50–100 µM	DOX 0.1–10 µM or CDDP 0.2–2 µg/mL for 24 h	↓ *p*AKT, ↑ caspase-3 ↓ IL-6/8 ↑ Osx, OPN, ALP, Col I, OCN	↓ proliferation ↑ apoptosis ↑ differentiation ↑ DOX/CDDP sensitivity	[286]
**Polydatin**	143B, MG-63	1–100 µM		↓ β-catenin signaling ↑ Bax/Bcl-2, caspase-3	↓ proliferation ↑ apoptosis	[303]
	MG-63	10–160 µM		↓ STAT3 signaling	↑ apoptosis, ↑ autophagy	[304,305]
**Polydatin**	SaOS-2/DOX, MG-63/DOX MG-63/DOX xenograft model	50–250 μM 150 mg/kg/d		↓ TUG1/Akt signaling	↓ proliferation ↑ apoptosis ↓ tumor growth	[306]
	U2OS, MG-63		paclitaxel		↓ proliferation ↓ migration ↑ cell cycle arrest	[304]
	SaOS-2	1–150 μM	ionizing radiation	↓ Wnt/β-catenin pathway ↑ lipid metabolite secretion	↑ differentiation ↑ cell cycle arrest ↑ radiation sensitivity	[307]
**Enterodiol, Enterolactone**	MG-63	0.1–10 mg/mL		↑ ALP activity ↑ ON, Col I	↓ proliferation ↑ differentiation	[308]

Strong evidence from breast, gastric, and prostate cancer cells subjected to combined treatment with resveratrol–DOX or resveratrol–CDDP has shown a synergistic behavior of resveratrol towards chemotherapeutic agents [309,310,311]. According to De Luca et al., the cotreatment of resveratrol with DOX and CDDP increased their cytotoxic effect on OS cells, suggesting that resveratrol might be a promising therapeutic adjuvant agent for OS cell treatment [286].

#### 7.1.2. Polydatin

Polydatin (3, 4′, 5-trihydroxystibene-3-β-mono-D-glucoside), also known as piceid, is a naturally occurring glucoside derivative of resveratrol, in which the glucoside group linked to position C-3 replaces the hydroxyl group [312,313] (Figure 5).

Polydatin was first extracted from the roots of *Polygonum cuspidatum* (Polygonaceae), which have a long history of use in traditional Chinese and Japanese medicines, but it also exists in a variety of other sources, including dietary plants such as grape, peanut, berries, and chocolate [313]. The *trans* form of polydatin is well known for its high therapeutic potential in a variety of medical domains, for example infection, inflammation, cardiovascular disorders, and aging-related diseases such as osteoporosis [314,315,316]. The Chinese FDA has approved polydatin for multiple phase II clinical trials, mainly for anti-shock applications [317]. 

Glucose substitution gives polydatin a more hydrophilic character than resveratrol, resulting in a significantly increased bioavailability and higher health-promoting/disease-modifying activities [318,319]. Furthermore, comparative studies of polydatin and resveratrol regarding antioxidative effects in vivo have revealed a better antioxidant activity of polydatin than resveratrol [312,320]. Polydatin has been recognized as a potent anticancer agent, with the ability to regulate various signaling pathways involved in the progression of several kinds of cancers [321,322]. It is mainly involved in cell cycle regulation, apoptosis, autophagy, signaling pathways, EMT, inhibition of inflammation and metastasis, and regulation of enzymes related to oxidative stress [323,324].

Moreover, recent studies by Mele et al. demonstrated that polydatin exerts a significant cytotoxic effect on cancer cells by Glucose-6-phosphate dehydrogenase inhibition, a rate-limiting enzyme in the pentose phosphate pathway which is altered in different malignant tumors [325]. A beneficial role of polydatin was also documented in prevention and treatment of OS. In an early study, Xu et al. evaluated the anti-OS activity of polydatin in human OS cell lines (i.e., 143B and MG-63). Polydatin dose-dependently inhibited proliferation by suppressing the β-catenin signaling and promoted apoptosis via upregulated expressions of Bax/Bcl-2 and caspase-3 in OS cells [303]. 

Polydatin also induced apoptosis via different mechanisms, such as reducing the expression/phosphorylation of STAT3 and increasing the expression of autophagy-related genes (Atg12, Atg14, BECN1, PIC3K3), thereby triggering autophagic cell death in MG-63 cells [305]. Further studies reported the efficacy of polydatin in drug-resistant models of OS [304,306]. The therapeutic effect of polydatin against DOX-resistant OS, in vitro and in a MG-63/DOX xenograft model, occurs via the oncogene taurine-upregulated gene 1 (TUG1)-mediated suppression of Akt signaling, which promotes apoptosis and prevents cell proliferation [306]. Additionally, polydatin enhances the chemosensitivity to the antineoplastic agent paclitaxel of U2OS and MG-63 cells and their paclitaxel-resistant variants, suppressing cell growth and migration and inducing cell cycle arrest in the S phase [304]. Interestingly, a recent study investigated the role accomplished by polydatin, alone or after radiation therapy, in the osteogenic differentiation of SaOS-2 and MG-63 [307]. 

As mentioned above, the osteogenic differentiation has a role in the pathogenesis of OS, considering that OS tumors deregulate the signaling pathways associated with osteogenic differentiation by arresting the cells as undifferentiated precursors [110]. 

In combination with radiotherapy, the pretreatment with polydatin promoted a radiosensitizing effect on OS cancer cells as demonstrated by the increased mineralization and osteogenic markers levels. The differentiation process was paralleled by the activation of the Wnt-β-catenin pathway and cell cycle arrest in the S phase. Additionally, the secretions of sphingolipid, ceramides, and their metabolites were analyzed using mass spectrometry in OS-treated cells. In past years, evidence has accumulated demonstrating that 2′-hydroxy ceramide/sphingolipids have distinct biological functions to regulate various cellular processes and cell differentiation by binding to specific target proteins [326]. 

Moreover, Sphingosine-1-phosphate has been reported to inhibit osteoclast formation and mineralization [327]. MS analysis demonstrated that polydatin-induced osteogenic differentiation was mediated by an increased expression and secretion of ceramides and sphingolipids and pretreatment with polydatin sensitized OS cells to ionizing radiation, suggesting that polydatin, in combination with radiotherapy, can consolidate the response to therapy of OS cells [307]. 

Various novel drug delivery systems, including nanoparticles [328], liposomes [329], micelles [330], quantum dots [331], and polymeric nanocapsules [332], have been designed to enhance polydatin pharmacodynamics and pharmacokinetics [333]. Polycaprolactone (PCL) is a biodegradable hydrophobic polyester used to obtain clinically applicable implantable nanostructures [334]. In their study, Lama and coworkers demonstrated that PCL nanofibers complexed to polydatin supported adhesion and promoted osteogenic differentiation in both SaOS-2 cells and bone MSCs, providing evidence of the osteogenic capacity of polydatin to create a biomimetic, innovative, and patented scaffold for both anticancer and regenerative purposes [335].

### 7.2. Lignans

In contrast to other types of phytoestrogens, lignan-type phytoestrogens have rarely been studied, although they are widely distributed in the plant kingdom. Cereals, grains, berries, and garlic are good dietary sources of lignans [30]. The plant lignans may occur in the form of aglycones and glycosides. After consumption, they can be converted by gut bacteria to form enterolignans (enterodiol and enterolactone), also known as “mammalian lignans”, which have a variety of biological activities, including tissue-specific ER activation and anti-inflammatory and apoptotic effects, that may influence disease risk in humans [336]. 

The effects of enterodiol and enterolactone on the viability and differentiation of MG-63 cells have been examined. Data suggested that both enterolactone and enterodiol have biphasic effects on cell proliferation, ALP activity, and transcriptional levels of osteonectin (ON) and Col I, showing induction at low doses and inhibition at high doses. 

The dose-dependent effects have been linked to the estrogenic and antiestrogenic properties of estrogen-like molecules as well as their ability to induce multiple signaling transduction [308]. 

## 8. Conclusions and Perspectives

OS is a disease of multifactorial origin which involves a complex interaction between a wide variety of factors and mechanisms that, when acting together, promote the deregulation of cellular signaling pathways, causing disturbances in bone tissue homeostasis [81,82]. Though rare, OS is the prevalent form of bone cancer among children and young adults. Despite advancements in treatment modalities, the prognosis for OS patients remains unfavorable, particularly in cases of metastasis and resistance to conventional therapies [5,6]. With the aim of improving patient outcomes, researchers have turned their attention to exploring new second-line treatment options (i.e., complementary and integrative medicine). 

Evidence collected from the literature indicates that phytoestrogens hold promise as a complementary therapeutic approach for OS, making them attractive candidates for further exploration. Mechanistically, phytoestrogens can interfere with genomic and non-genomic signaling pathways, such as NF-kB, PI3K/Akt, or MAPK/ERK, control cell cycle progression, initiate apoptosis events, and inhibit angiogenesis and metastasis. Several reports have also highlighted their ability to enhance the efficacy of chemotherapy agents and overcome drug resistance. Among them, genistein, resveratrol, and quercetin have the most evidence of effectiveness [169,171,181,186,187,286,306].

Despite the abundance of in vitro studies, the role and involvement of ERα and/or ERβ in the phytoestrogen-dependent modulation of OS cells has been only partially investigated to date, although the estrogen signaling may be involved in the growth and progression of this malignancy [14,122,124]. 

To fully assess the impact of these compounds on OS initiation, progression, and development, the Erα/ERβ ratio in specific OS models should also be considered. This is important due to the considerable variation in ER-mediated cellular response based on cell type, along with the selectivity and concentration of phytoestrogens. Taken together, these factors play a crucial role in determining the multifaced effects exerted by these plant estrogenic molecules on cancer cells [48,49,50]. 

Although in vitro and preclinical studies have shown promising results, it is crucial to evaluate the true potential of phytoestrogens as a viable option for OS therapy through the rigorous assessment of their efficacy, safety, optimal dosage, and treatment duration in clinical settings. In this regard, preliminary clinical studies have been conducted to investigate the pharmacokinetics of compounds derived from plant sources, including phytoestrogens, in individuals with premalignancies or patients with different types of cancer [79,197,337]. Among the various molecules studied, only clinical investigations on green tea extracts have consistently yielded results, supporting their potential for chemoprevention in premalignant cervical disease, prostate cancer, and leukoplakia. Moreover, existing data generally supports the safety of small doses of purified phytoestrogen consumption as a medication for breast cancer [215]. Conversely, further clinical trials are needed to sustain the role of soy isoflavones in preventing cancer development and progression [215,337]. Overall, the protective effect of phytoestrogens against cancer development remains a topic of ongoing debate [79]. This controversy stems primarily from a notable discrepancy between the outcomes of mechanistic studies conducted in vitro and the observations made in clinical settings. The main reason for these inconsistencies is the frequent use of non-physiological concentrations of phytoestrogens in mechanistic studies, which are difficult to achieve through normal dietary intake [79,338].

Interestingly, some ongoing research is focusing on the development of highly specific and long-acting analogues or drug delivery strategies to improve the pharmacodynamics and pharmacokinetics of phytoestrogens in human OS models [265,276,328,332,335]. This may encourage the exploitation of the therapeutic potential of phytoestrogens in managing OS and other types of cancers. 

## Figures and Tables

**Figure 1 ijms-24-13344-f001:**
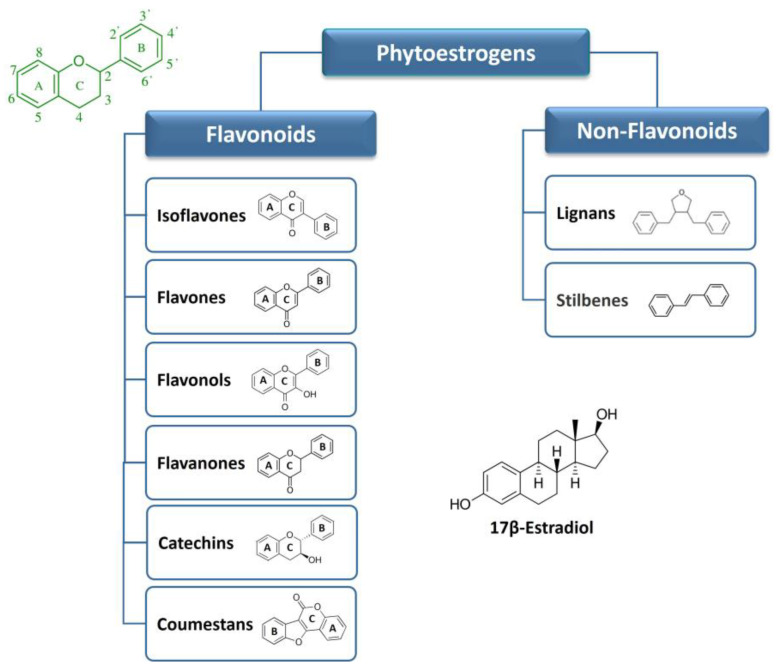
Basic chemical structures of the major classes of phytoestrogens. The different types of phytoestrogens share a structural similarity with 17β-Estradiol. In green, the flavonoid skeleton showing rings A, B, and C and the numbering.

**Figure 2 ijms-24-13344-f002:**
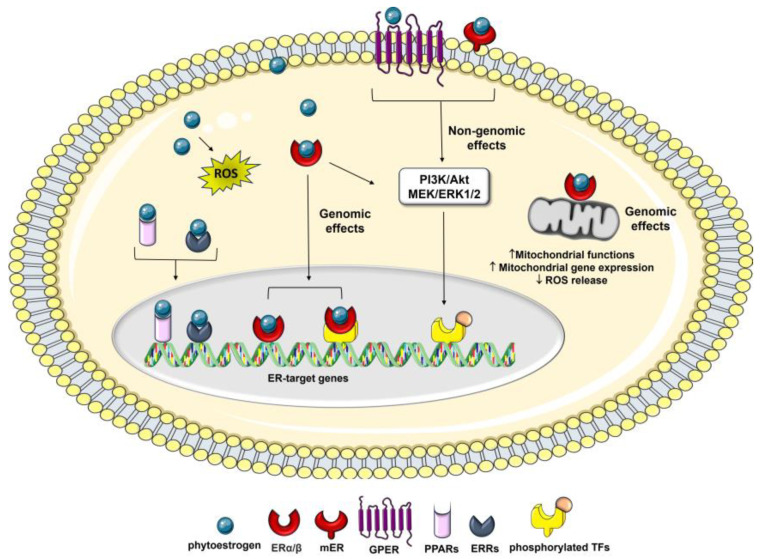
Intracellular mediators of the effects of phytoestrogens on cancer cells. Phytoestrogen can interact with the two types of ER, the intracellular ERα and Erβ, and membrane-associated mERs and GPER, activating downstream genomic and non-genomic effects which ultimately affect cell cancer phenotypes. The genomic pathway can involve ER interactions with other transcription factors (TFs), such as CREB, AP-1, Sp1, and NF-κB. Phytoestrogens can also act through ER-independent mechanisms which induce oxidative stress-mediated signaling by generating ROS, as well as interacting with other nuclear receptors, such as PPARs and ERRα/γ.

**Figure 3 ijms-24-13344-f003:**
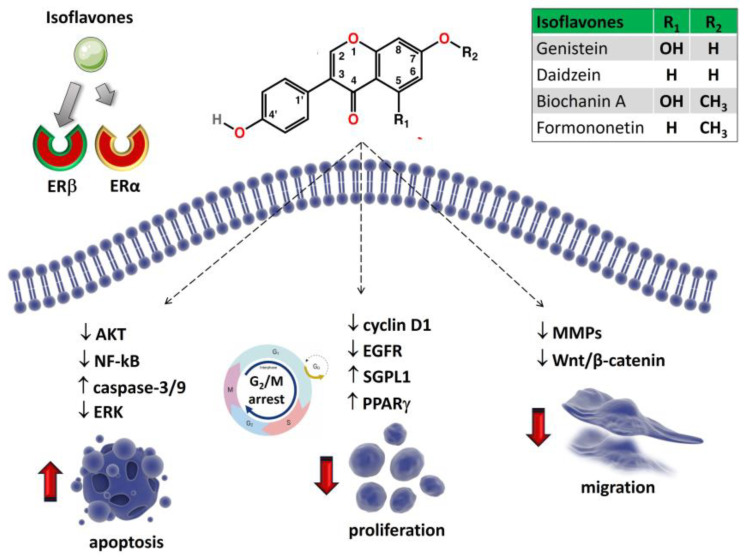
Chemical structure of the most common isoflavones and some of their possible targets in OS cells. Isoflavones, especially genistein, bind ERβ with a significant higher affinity than ERα. Downward arrows represent downregulation or reduction. Upward arrows represent upregulation or increase.

**Figure 4 ijms-24-13344-f004:**
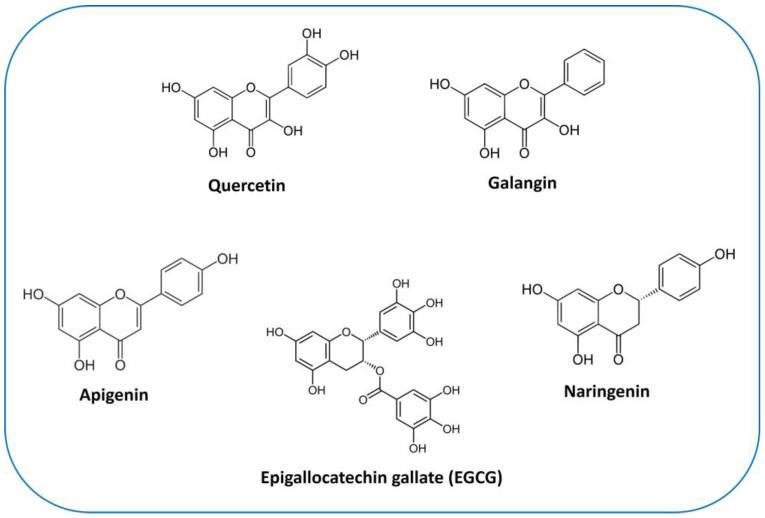
Chemical structure of flavonoids discussed throughout the text.

**Figure 5 ijms-24-13344-f005:**
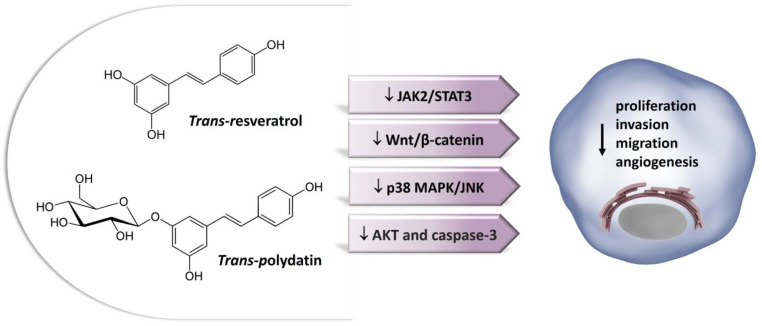
Downregulation of various pathways involved in the anti-OS activity of the stilbene resveratrol, and its glycosidic form polydatin.

**Table 1 ijms-24-13344-t001:** Relative binding affinities of various dietary phytoestrogens for ERα and ERβ, with values for E2 arbitrarily set as 100. The phytoestrogens reported are those discussed in Section 6 and Section 7.

Phytoestrogen	Relative Binding Affinity		References
	ERα	ERβ	
*Isoflavones*			
Genistein	4	87	
Daidzein	0.1	0.5	[19]
Biochanin A	<0.01	<0.01	
Formononetin	<0.01	<0.01	
*Flavonols*			
Quercetin	0.01	0.04	[19]
Galangin	ND	ND	
*Flavones*			
Apigenin	0.3	0.6	[19]
*Flavanones*			
Naringenin	0.01	0.11	[19]
*Stilbenes*			
Resveratrol	6.11–11.2 ^a^	4.7–15.66 ^a^	[52]
Polydatin	ND	ND	

^a^ values reported by Bowers et al. [52] were obtained using a different methodology.

**Table 2 ijms-24-13344-t002:** Effects of flavonoids on osteosarcoma. Downward arrows represent downregulation or reduction. Upward arrows represent upregulation or increase.

Phytoestrogen	Cell Line/ In Vivo Model	Concentrations	Combined Treatment	Molecular Mechanism	Observed Effects	References
**Genistein**	U2OS	1 µM		ER-mediated downregulation of EGFR	↑ differentiation ↑ apoptosis ↑ cell cycle arrest	[154]
	MG-63	2.5–30 μmol/L		↑ matrix vesicles, ↑ ALP activity	↑ differentiation ↑ mineralized bone noduli ↓ proliferation	[158]
	MG-63, SaOS-2	10–30 µM		↓ PTK, ↑ ER	↓ synthesis and secretion of GAGs/PGs	[162]
	MG-63	40–80 µM		↑ PPARγ pathway	↑ cell cycle arrest ↓ proliferation	[164]
	SaOS-2, MG-63	1–100 µM	calcitriol 10 nM for 48 h	↑ SGPL1, VDR, ERβ	↑ calcitriol sensitivity ↓ proliferation ↓ extracellular acidification ↓ mitochondrial respiration	[165]
	MG-63, U2OS	10–100 μmol/L	gemcitabine 0.5 µmol/L for 72 h	↓ Akt/NF-κB pathway	↑ gemcitabine sensitivity ↑ apoptosis	[169]
**Daidzein**	U2OS	1 µM		ER-mediated downregulation of EGFR	↑ differentiation ↑ apoptosis ↑cell cycle arrest	[154]
	143B, U2OS xenograft mouse model	10–500 μmol/L 20 mg/kg every 2 days		↓ Src-ERK pathway	↑ apoptosis ↑ cell cycle arrest ↓ migration ↓ tumor weights	[172]
**Biochanin A**	MG-63, U2OS	5–30 μg/mL	DOX 1 μg/mL for 24 h	↑ caspase-3/9, ↑ Bax: Bcl-2/Bcl-XL ratio	↑ DOX sensitivity ↓ proliferation ↑ apoptosis	[173]
	MG-63, U2OS	5–80 μM		↓ PCNA, cyclin D1, Bcl-2, ↑ Bax, caspase-3; ↓ MMP-9, N-cadherin, ↑ E-cadherin	↑ cell cycle arrest ↑ apoptosis ↓ migration, invasion	[174]
**Formononetin**	U2OS, tumor-bearing nude mice	20–80 μM, 80 mg/kg/d		↓ Bcl-2, miR-375, ↑ Bax	↑ apoptosis ↓ tumor weights	[175]
	U2OS	5–100 μM		↑ caspase-3 and Bax, ↓ Bcl-2 ↓ ERK and PI3K/AKT pathway	↑ apoptosis	[176]
	MG-63	15–45 μM		↑ miR-214-3p/PTEN pathway	↓ proliferation ↑ apoptosis	[177]
	tumor-bearing nude mice	25–100 mg/kg/d		↓ ESR1, p53, ERBB2	↓ tumor weights	[178]
**Quercetin**	MG-63	20–320 μM		↑ cytochrome C, caspase-3/9, Bax, ↓ Bcl-2	↑ apoptosis	[179]
	HOS, ATCC 1543	10–1000 µM		↓ cyclin D1, ↑ caspase-3, cleaved PARP	↑ cell cycle arrest ↓ proliferation ↑ apoptosis	[180]
	U2OS/MTX300	10–50 µM		↑ cytochrome C, caspase-3, Bax, cleaved PARP; ↓ Bcl-2, Akt	↑ apoptosis ↓ proliferation	[181]
	143B	10–100 µM		↑ caspase-3, cleaved PARP	↑ cell cycle arrest ↑ apoptosis ↓ adhesion ↓ migration	[182]
	MG-63	50–200 µM		↑ LC3B-II/LC3B-I ratio, ↓ ROS-NUPR1 pathway	↑ autophagy	[183]
	HOS, MG-63, tumor-bearing nude mice	20–100 µM 25–100 mg/kg/d		↓ HIF-1α, VEGF, MMP-2/9	↓ migration, invasion ↓ tumor growth	[184]
	U2OS, SaOS-2			↓ PTHR1	↓ proliferation ↓ adhesion ↓ migration, invasion	[185]
	143B	5 μM	CDDP 5 μM for 24 h	↑ miR-217- KRAS axis	↑ CDDP sensitivity ↓ proliferation ↓ migration, invasion	[186]
	SaOS-2	10–200 μM	MTX 10–200 μMfor 48 h	↑ p53, CBX7, CYLD, ↓ Bcl-2, miR-223	↑ MTX sensitivity ↓ proliferation ↑ apoptosis	[187]
**Galangin**	MG-63, U2OS	5–300 µM		↓ PI3K/Akt pathway ↓ cyclin D1, MMP-2/9, ↑ p27Kip1, caspase-3/8	↓ proliferation ↑ apoptosis ↓ invasion	[188]
	MG-63, U2OS tumor xenograft mouse	25–100 μM 50, 100 mg/kg/d		↑ TGF-β1/Smad2/3 pathway ↑ Col I, ALP, OPN, OCN	↑ differentiation ↓ tumor growth	[189]
**Apigenin**	U2OS tumor xenograft mouse	50–200 μM 2 mg/kg every 3 days		↑ caspase-3/8/9, BAX, AIF	↑ apoptosis ↓ tumor growth	[190]
	U2OS, MG-63	20–100 μg/mL		↓ Wnt/β-catenin pathway	↓ proliferation ↓ invasion	[191]
**Naringenin**	HOS, U2OS	100–500 μM		↑ ROS-Mediated ER Stress	↑ autophagy ↑ apoptosis	[192]
**EGCG**	MG-63, 143B, SaOS-2 tumor-bearing nude mice	10–50 μM 10–40 mg/kg every 2 days		↓ Wnt/β-catenin pathway	↓ proliferation ↓ migration, invasion ↓ tumor growth	[193]
	MG-63, U2OS tumor-bearing nude mice	0.0125–0.1 g/L 30 mg/kg/d		↑ miR-1	↓ proliferation ↓ tumor growth	[194]
	U2OS	5–50 μM	IL-1Ra1 ng/mL	↓ MMP-2, VEGF ↓ IL-6/8	↓ proliferation ↓ invasion	[195]
	U2OS, SaOS-2	20 μg/mL	DOX 1–2.5 μM	↓ SOX2OT	↑ autophagy	[196] (p. 7)

## Data Availability

Not applicable.

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
