# Peer review of "Multi-Anticancer Activities of Phytoestrogens in Human Osteosarcoma"

_ijms, 2023, doi:10.3390/ijms241713344_

Round 1
Reviewer 1 Report
In their review, Cimmino et al. describe the activities of phytoestrogens in osteosarcoma. For this, they first described the structure of these molecules and their classifications, then they presented their known molecular functions in cancers with a special focus on osteosarcoma, and finally described listed the functions of flavonoids and non-flavonoids in osteosarcoma.
This review is extremely detailed and very well written. Figures and tables are also very informative and well distributed throughout the text. No addition or modification needed.
Author Response
We are grateful to Reviewer 1 for his/her positive comments and for considering our manuscript valuable and suitable for publication in its original form.
Reviewer 2 Report
The Authors review on the multi-anticancer activities of phytoestrogens in human osteosarcoma (OS) which provides an extensive overview of current literature on the effects of phytoestrogens on human OS models. It delves into the multiple mechanisms through which 18 of these molecules regulate the cell cycle, apoptosis and key pathways implicated in the growth and progression of OS. As positive point, it includes biologics not usually listed in other papers already in the literature.
Page12-Page 13, the review about anti-OS of Daidzein, Biochanin A, did not mention the content of ER (ERbeta and ERalpha)
Page 14-Page 16, most of the content of Quercetin and formononetin were not associated with ER (ERbeta and ERalpha). The role of ER (ERbeta and ERalpha) in the modulation of quercetin on the suppression of proliferation, cell cycle arrest, induction of apoptosis and autophagy should be summarized.
